# HANDLING DELAY IN REAL-TIME REINFORCEMENT LEARNING

**Ivan Anokin**[12], **Rishav Rishav**[13], **Matthew Riemer**[124], **Stephen Chung**[5]
**Irina Rish**[126], **Samira Ebrahimi Kahou**[136]
[1]Mila  [2]Université de Montréal  [3]University of Calgary  [4]IBM Research
[5]University of Cambridge  [6]CIFAR AI Chair
`ivan.anokhin@mila.quebec`

## ABSTRACT

Real-time reinforcement learning (RL) introduces several challenges. First, policies are constrained to a fixed number of actions per second due to hardware limitations. Second, the environment may change while the network is still computing an action, leading to *observational delay*. The first issue can partly be addressed with pipelining, leading to higher throughput and potentially better policies. However, the second issue remains: if each neuron operates in parallel with an execution time of $\tau$, an $N$-layer feed-forward network experiences observation delay of $\tau N$. Reducing the number of layers can decrease this delay, but at the cost of the network's expressivity. In this work, we explore the trade-off between minimizing delay and network's expressivity. We present a theoretically motivated solution that leverages *temporal skip connections* combined with history-augmented observations. We evaluate several architectures and show that those incorporating temporal skip connections achieve strong performance across various *neuron execution times*, reinforcement learning algorithms, and environments, including four Mujoco tasks and all MinAtar games. Moreover, we demonstrate parallel neuron computation can accelerate inference by 6-350% on standard hardware. Our investigation into temporal skip connections and parallel computations paves the way for more efficient RL agents in real-time setting.

## 1 INTRODUCTION

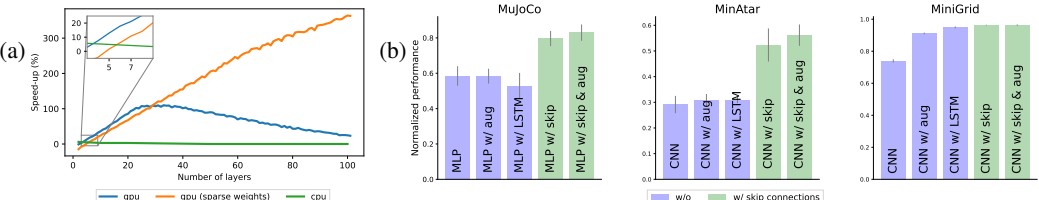

Figure 1: **(a)** Parallel computations of layers speed-up inference time. Speed-up on GPU is achieved using default Pytorch software and widely accessible Nvidia GPU. **(b)** Normalized averaged performance and standard error of agents in parallel computation framework. Agents with skip connections and history-**aug**mented observations exhibit strong performance. Performance is averaged across the following environments: HalfCheetah-v4, Walker2d-v4, Ant-v4 and Hopper-v4 on Mujoco, all six environments on MinAtar, and Empty-Random-5x5-v0 and DoorKey-5x5-v0 on MiniGrid. Performance on Mujoco is also averaged across four different neuron execution times.

Neural network inference presents several challenges in real-time reinforcement learning (RL) as the environment can change significantly, even during the networks' inference process. One major challenge is that the inference time directly impacts throughput, the number of actions the agent can produce per second. High throughput is important in domains like robotics, algorithmic trading, and real-time gaming, where frequent decision-making can significantly improve policy performance.

To address this challenge, a straightforward approach is to speed up inference by employing pipelining techniques. In a pipelined architecture, instead of waiting for the entire neural network to complete its forward pass on one input before processing the next, each layer begins processing the subsequent input as soon as it produces its output for the current one (Carreira et al., 2018; Iuzzolino et al., 2021). This approach increases the throughput of a neural network (see Fig. 1a), as layers are effectively working in parallel but on different inputs. Throughout this paper, we refer to this approach as the *parallel computation framework*.

However, even within the parallel computation framework, a traditional $N$-layer feed-forward neural network still suffers from another issue known as *observational delay*: the agent's action at time step $t$ is based on an observation from time step $t - N\delta$ where $\delta$ denotes execution time of each layer. This delay arises because, in a pipelined system, each layer is processing data from different time steps simultaneously – layer 1 processes input from time $t$, layer 2 processes the output of layer 1 from time $t - \delta$, and so on (as illustrated in the center graph of Fig. 2). This challenge leads us to the central question of this paper:

*If we use parallel computations of layers, how do we address observational delay?*

Reducing the number of layers can mitigate delay but limits the network's expressivity. To overcome this, we propose using *temporal skip connections*. Traditionally, skip connections are used to stabilize training and allow gradient flow in deep networks (Ronneberger et al., 2015; He et al., 2016). However, within the parallel computation framework, skip connections offer another advantage: they do not only shortcut between layers along depth, but also along time, by sending activations forward in time (see the rightmost graph in Fig. 2). This temporal application of skip connection in the parallel computation setting reduces the observational delay. Nevertheless, the computational paths through these temporal skip connections are shorter than those without them and thus offer limited expressivity compared to longer paths through more neurons.

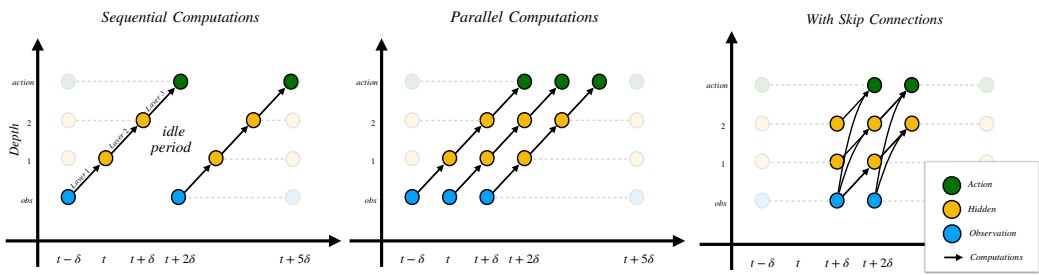

Figure 2: Computation flow of agents. Left graph represents sequential computations and the central graph – parallel computations of layers. $\delta$ is execution time of each neuron (or layer). All nodes at each column are available at the same time and can be processed further in parallel. The right architecture with skip connections exhibits less delay as it performs shortcuts along time-steps.

We explore the trade-off between delay and network expressivity and investigate various types of architectures to find an optimal balance. Our theoretical analysis quantifies the impact of skip connections on reducing the regret associated with observational delay. Furthermore, we justify the importance of augmenting observations with past data in architectures with temporal skip connections. Experiments confirm the importance of skip connections and history-augmented observation (see Fig. 1b), and our analysis shows that the skip connection offers a fast but less refined path for processing inputs, while the main connections provide a slower but more refined path. Our results show that in many environments this allows the policy in a parallel computation setting to achieve similar performance to an oracle agent with an instantaneous forward pass, provided the inference time of a layer is not large.

While the parallel layer computation approach and temporal skip connections were proposed before to accelerate predictions on image (Iuzzolino et al., 2021; Fischer et al., 2018) and video (Carreira et al., 2018; Kugele et al., 2020) domains, this is the first application in RL – a domain where one bad action can critically impact the entire trajectory due to the agent's influence on the environment.

To summarize, we introduce a solution to real-time RL: speeding up inference time by parallel computations of layers and addressing associated observational delay. We demonstrate that parallel computations significantly improve throughput on modern hardware like GPUs. To address the observational delay, we provide a theoretically justified solution using temporal skip connections and history-augmented observations. Our experiments demonstrated its effectiveness across various cases, paving the way for more efficient RL agents in real-time setting.

## 2 RELATED WORK

**Parallel computations of neurons (or layers).** Parallel processing of information is consistent with popular mathematical models of the human cortex (Tomita et al., 1999; Betti & Gori, 2019; Kubilius et al., 2018; Larkum, 2013), where neurons operate asynchronously. Inspired by this, several attempts have been made to parallelize neural networks, aiming to maximize processing resource utilization and reduce latency. Carreira et al. (2018) introduced parallel video networks that employ parallel layer computations and temporal skip connections, significantly boosting throughput (or frame rate) during inference. Similarly, Iuzzolino et al. (2021) explored this approach for still images, enabling fast "anytime predictions" that improve over time. Additionally, Fischer et al. (2018) provided a theoretical framework for these ideas, and Kugele et al. (2020) applied them for Spiking Neural Networks on image and video domains. Unlike these approaches, we apply these ideas in RL.

Several studies have proposed techniques to handle parallel computations of layers not only during the forward pass but also during the backward pass (by modifying or replacing backpropagation) in both training and inference. Sideways (Malinowski et al., 2020; 2021) achieved this with approximate backpropagation in the video domain. Asynchronous Coagent Networks (Kostas et al., 2020) and Chung (2022) introduced methods where each neural network unit operates independently to maximize its own reward, enabling asynchronous inference and training of neurons. However, Sideways focuses on video data, and both Coagent Networks and Chung (2022) are limited to a small number of neurons, making scalability to larger networks challenging compared to our approach.

**Delay in RL.** Early works on handling delays in traditional RL settings include (Walsh et al., 2007; Bander & White, 1999; Katsikopoulos & Engelbrecht, 2003; Altman & Nain, 1992). Notably, (Katsikopoulos & Engelbrecht, 2003) was the first to introduce the notion of a Delayed Markov Decision Process (DMDP). However, their results have not been fully translated into Deep RL.

Recent efforts have addressed delays in Deep RL. Firoiu et al. (2018) tackled delay by predicting future observations, while Wang et al. (2023) trained the critic without delay, augmented state information with historical data, and used self-supervised losses to improve performance on DMDPs. The RLRD method (Bouteiller et al., 2021) further enhanced the critic by augmenting its input with future on-policy actions available due to delay, resulting in more accurate value estimations.

These approaches consider delay as an external factor to the agent. However, our agent inherently introduces delays due to parallel computations, resulting in additional interplay between the agent's architecture and these inherent delays. This allows us to introduce more inductive biases, such as temporal skip connections, into the neural network architecture to effectively mitigate such delays.

## 3 PROBLEM SETTING AND NOTATION

A Markov Decision Process (MDP) (Puterman, 1994; Sutton & Barto, 2018) is defined as a tuple $(\mathcal{S}, \mathcal{A}, \mathcal{P}_0, \mathcal{P}, \gamma)$, where $\mathcal{S}$ and $\mathcal{A}$ are the state and action spaces, respectively. $\mathcal{P}_0$ specifies the initial state distribution such that $\mathcal{P}_0(s)$ is the probability of a state $s \in \mathcal{S}$ being an initial state. $\mathcal{P}$ specifies the state transition probability such that $\mathcal{P}(s', r|s, a)$ is the probability of reaching to a new state $s' \in \mathcal{S}$ with an immediate reward $r \in \mathbb{R}$ after taking an action $a \in \mathcal{A}$ at a state $s \in \mathcal{S}$. $\gamma \in [0, 1)$ is the discount factor, which weights the importance of rewards at future steps. It is typically assumed that MDPs are "pausable" i.e. that the agent and environment proceed in a turn-based interaction framework where each waits for each other before proceeding. In realtime environments, however, the agent and environment each proceed at their own pace (Travnik et al., 2018).

Then we define an asynchronous delayed MDP as a tuple $(\mathcal{S}, \mathcal{A}, \mathcal{P}_0, \mathcal{P}^d, \gamma, \beta, d)$, which extends the standard notion of an MDP by defining $\beta$ – the default behavior policy of the system between actions taken by the agent, $d \in \mathbb{N}$ is amount of delay, and fixed interaction frequency, indicating the number of

environment steps between the agent's actions at the same time. $\mathcal{P}^d$ is the delayed transition probability function, which we will define to model the environment's dynamics under the influence of both the agent and the default policy. $\mathcal{P}^d(s', R \mid s, a) = \mathbb{E}_\beta \left[ \prod_{k=1}^d \mathcal{P}(s_k, r_k \mid s_{k-1}, a_{k-1}) \mid s_0 = s, a_0 = a \right]$ where $s_0 = s, a_0 = a, s_d = s'$ and $R = \sum_{k=1}^d r_k$ is the cumulative reward over $d$ steps. The delayed transition probability function $\mathcal{P}^d$ captures the probability of transitioning from state $s$ to state $s'$ over $d$ steps, starting with the agent's action $a$ and followed by the default policy $\beta$.

We extend asynchronous delayed MDP further to asynchronous delayed observation MDP (asynchronous DOMDP) to define that agent observes history of past states $(s_{t-dN}, s_{t-d(N-1)}, \ldots, s_{t-d})$ where $N$ will define number of layers in neural network later.

## 3.1 FORMALIZING THE PARALLEL COMPUTATION FRAMEWORK

We execute layers of our neural network in parallel to speed-up inference in realtime settings. Thus, we need to incorporate computational constraints related to the parallel computations. We define $\delta$ as *neural execution time* i.e. the number of environment steps that pass during the computation of a single neural network layer. If $\delta > 1$, a default policy $\beta$ takes control for $\delta$ steps. For $\delta < 1$, we either accelerate the environment or group $\lceil 1/\delta \rceil$ layers together to form a new macro layer. The agent's policy, $\pi$, represented with $N$-layer neural network, observes a history of past states, $h_\delta$, at intervals of $\delta$: $\pi(h_\delta) = \pi \left( s_{t-\lceil N\delta \rceil}, \ldots, s_{t-\lceil 2\delta \rceil}, s_{t-\lceil \delta \rceil} \right)$. The policy must respect the computational constraint that it cannot process past state $s_{t-k}$ through more than $\lfloor k/\delta \rfloor$ layers before producing action $a_t$. As such, our goal is to find a policy $\pi(h_\delta)$ that maximizes cumulative rewards in a asynchronous DOMDP with delay of $\lceil \delta \rceil$, subject to the constraint that $L(s_{t-k}, a) \leq \lfloor k/\delta \rfloor \ \forall k \in \{\lceil \delta \rceil, \ldots, \lceil N\delta \rceil\}$ where $L(s_{t-k}, a)$ is number of layers between $s_{t-k}$ and $a$. We can view a neural network as a directed acyclic graph (DAG), where the nodes represent input data or intermediate computational results, and the edges represent the computational operations. $L(s, a)$ is the path in this graph from $s$ to $a$. We define a neural network that only consists of the longest paths in $\pi(h_\delta)$ as a *vanilla feed-forward neural network*. All other paths will be referred to as *temporal skip connections*.

## 3.2 SOURCES OF REALTIME REGRET

We define a delay regret as the difference in performance of the optimal policy in the original MDP and the optimal policy in asynchronous DOMDP. Similarly, an inaction regret is defined as the difference in performance of the optimal policy in the original MDP and performance of default policy $\beta$ in asynchronous DOMDP. We give the formal definitions in Appendix F.

## 4 METHOD

We show benefits of temporal skip connections for minimization delay regret bound $\Delta_{\text{delay}}$ in Proposition 1 and benefits of temporal skip connections combined with the state augmented with recent actions in Proposition 2.

## 4.1 ADDRESSING DELAY

In a vanilla feedforward neural network deployed to address realtime RL, actions $a_t$ are based on states $s_{t-N\delta}$ delayed by $N\delta$ steps. Temporal skip connections directly alleviate this issue as actions $a_t$ are now based on a set of $N$ states $\{s_{t-N\delta}, \ldots, s_{t-\delta}\}$. As a result, skip connections lead to a tighter lower bound on delay regret, $\Delta_{\text{delay}}(t)$ in a worst case environment.

**Proposition 1** *(Tighter Delay Regret Bound): For any vanilla $N$ layer neural network without temporal skip connections in parallel computation framework, the regret resulting from delay $\Delta_{delay}^{vanilla}(t)$ after $t$ steps in a worst case environment can be lower bounded by:*

$$\Delta_{delay}^{vanilla}(t) \in \Omega(t(1 - (p_{minimax})^{\lceil N\delta \rceil})) \tag{1}$$

*where $p_{minimax} := \min_{s \in \mathcal{S}, a \in \mathcal{A}} \max_{s' \in \mathcal{S}} p(s'|s, a)$ is a measure of environment stochasticity. However, a network with temporal skip connections achieves a tighter bound on delay regret $\Delta_{delay}^{skip}(t)$:*

$$\Delta_{delay}^{skip}(t) \in \Omega(t(1 - (p_{minimax})^{\lceil \delta \rceil}))  \tag{2}$$

*which is less sensitive to the environment stochasticity measured by $p_{minimax}$.*

Following the lower bound on delay regret established in (Riemer et al., 2024), the delay regret depends on the number of stochastic environment steps between an action and the input used to produce it. Temporal skip connections enable the policy to incorporate the state from $\lceil \delta \rceil$ steps ago, whereas a vanilla feedforward network can only condition on steps $\lceil N\delta \rceil$ in the past. This can lead to an exponential reduction in the policy's inaccuracy caused by the stochasticity in the environment, which becomes especially prominent for environments that are highly stochastic or neural networks with a large number of layers.

## 4.2 Addressing Training Stability

Another difficulty with vanilla feedforward neural networks, even with parallel inference, is that the effective delayed decision process where actions $a_t$ are taken based on the delayed state $s_{t-N\delta}$ is non-Markovian. This fact will lead to unstable learning in many environments as typical RL algorithms are not expected to converge in this regime. Meanwhile, this is another key issue that can be addressed with temporal skip connections and augmenting state with recent actions. With this architecture, we have access to all previous actions when computing $a_t$ and thus can consider a stable augmented state space $\tilde{s}_t = (s_{t-N\delta}, a_{t-N\delta:t-1})$ that the decision process is Markovian with respect to as $p(r_t, \tilde{s}_{t+1}|\tilde{s}_t, a_t)$ is stationary and stable over time.

**Proposition 2** *(Markovian Property): A vanilla $N$ layer neural network without skip connections in parallel computation framework bases its actions on the delayed state $s_{t-N\delta}$ and experiences non-Markovian environment transitions $p(r_t, s_{t+1}|s_{t-N\delta}, a_t)$ without having access to $a_{t-N\delta:t-1} = a_{t-N\delta}, ..., a_{t-1}$. These actions are available when using temporal skip connections, making environment Markovian based on the augmented delayed state space $\tilde{s}_t = (s_{t-N\delta}, a_{t-N\delta:t-1})$.*

The vanilla network is non-Markovian as it depends on past actions from a changing policy. Skip connections and past actions remove this non-stationary dependency. This property can be illustrated with the following example: If the action $a_t$ at time $t$ is based on the state $s_{t-1}$, the transition probability function becomes $P(s'|s_{t-1}, a_t) = P(s'|s_t, a_t)P(s_t|s_{t-1}, a_{t-1})\pi(a_{t-1}|s_{t-2})$. While $P(s'|s_t, a_t)$ is stationary, the term $\pi(a_{t-1}|s_{t-2})$ is non-stationary because the policy changes throughout learning. However, by augmenting the state with $a_{t-1}$, the policy term disappears, and the transition function becomes stationary. Proposition 2 is an important point to emphasize as it extends Proposition 1 to explain optimization issues related to delay that may be present even when the environment is deterministic within the parallel computation framework. When using an earlier state to generate a policy, the effect of the actions of that policy also depend on the actions taken between action computations because of the non-Markovian nature of that input representation. As such, *the transition dynamics appear nonstationary as the policy itself changes and appear stochastic when the policy is stochastic.* This serves to slow down learning and leads to instability that hurts sample efficiency as we demonstrate in our experiments.

When using temporal skip connections, our policy conditions on $N$ previous states and actions while only the most recent one $s_{t-\delta}$ and $a_{t-\delta}$ are needed in our derivations of Propositions 1 and 2. However, utilizing these previous states is still helpful within the framework of parallel layer computation because we are able to consider more neural network layers for states that are more outdated. This way the policy can be more expressive with respect to previous states than it is to the most recent state. This is a useful feature in environments that are relatively stable and predictable across each step while requiring complex high-level reasoning. For example, in a maze environment the overall structure of the maze may stay constant across steps, so even distant steps can be useful in processing a higher level plan of action with more recent steps being used to encode the representation of the agent's current location. Indeed, our experiments validate the value of adding more layers even with outdated states in delayed variants of popular environments within the deep RL community.

**Performance gap.** Propositions 1, 2 highlight the performance gap between agents with and without skip connections and last-action augmentation, in terms of delay regret. Besides, Propositions 1 and 3 (Appendix G) provide insights into the performance gap between the instantaneous and real-time

actors in a parallel computation framework under worst-case environments. Notably, even with skip connections, the delay $\delta$ remains. In contrast, the instantaneous actor does not experience any delay or inaction regrets. The closer the environment is to a worst-case scenario, the more pronounced the performance gap becomes.

However, when using skip connections, state-augmentation with last actions, deterministic environment, and neural execution time less than 1, Propositions 1, 3 falls short to differentiate between instantaneous and real-time actors. In this case, the real-time actor also exhibits zero delay and inaction regrets. Nevertheless, we anticipate the real-time actor to perform worse than the instantaneous actor. Skip connections may lack the expressivity needed to efficiently differentiate between distinct environment states, effectively perceiving the environment as stochastic. This limitation makes Proposition 1 relevant again.

### 4.3 ALGORITHM

We apply Soft Actor Critic (SAC) (Haarnoja et al., 2018) for continuous action-space environments or PPO for discrete [1] . We train a critic without delay following suggestions from (Wang et al., 2023) and an actor with appropriate delay and restriction following Subsection 3.1 with vanilla backpropagation. We employ last action repetition as default policy, $\beta$, if $\delta > 1$.

The basic structure of our SAC algorithm is presented in Algorithm 1. To begin collecting experience, we initialize the first observation from the environment and set initial hidden activations, depicted in Fig. 2, to zero [2]. While the critic is trained online without delay, our actor is trained within the parallel computation framework by unrolling on sub-trajectories sampled from the buffer (with hidden activation set to zero at the first state of a sub-trajectory), allowing all weights to be available for backpropagation. For details on the PPO variant of the algorithm, refer to Appendix A.

---

**Algorithm 1** Soft Actor-Critic Algorithm with parallel neuron computation.

---

1: Init an actor and a critic with random parameters.
2: Set initial state to be $s_0, h_0^0, ..., h_0^N$ , where $h_0^j$ is activations for layer $j$ at a time step $0$.
3: Wrap the environment with sticky actions or repeating observations wrapper if needed based on neural execution time.
4: **for** $t \in 0, \ldots, L$ **do**
5: $\quad a_t, h_{t+1}^0, ..., h_{t+1}^N = Actor(s_t, h_t^0, ..., h_t^N)$ (Query current policy for the next action and next Actor's hidden activations given current observation and hidden activations)
6: $\quad$ Take the action $a_t$ and receive $\{r_t, s_{t+1}\}$ from the environment.
7: $\quad$ Put $\{s_t, a_t, r_t, s_{t+1}\}$ to the buffer.
8: $\quad$ Sample transition $\{s_i, a_i, r_i, s_{i+1}\}$ from the buffer and update the critic on it.
9: $\quad$ Sample sub-trajectory from the buffer $\{s_i, a_i, r_i, s_{i+1}, ..., r_{i+k}, s_{i+k}\}$
10: $\quad$ Init $h_0^0, ..., h_0^N$ and simulate the actor dynamic forward on given sub-trajectory.
11: $\quad$ Update the actor on the last transition of the sub-trajectory (via back-propagation through time if needed)
12: **end for**

---

## 5 EXPERIMENTS

We perform our main experiments on Mujoco (Todorov et al., 2012), MiniAtar (Young & Tian, 2019) and MiniGrid (Chevalier-Boisvert et al., 2023) environments. Mujoco has a continuous action space, while MiniAtar and MiniGrid have discrete action spaces. We train our agents using SAC for Mujoco and PPO (Schulman et al., 2017) for MiniGrid and MinAtar. We report mean and standard error (SE) in all our plots and experiments unless stated otherwise. We normalize return for every environment and neuron execution time with respect to vanilla SAC or PPO performance without delay. Additional architectural and training details can be found in Appendix E.

---

[1]Full code is available at `https://github.com/avecplezir/realtime-agent`.
[2]Since we initialize the hidden activations to zero, the first delayed actions can be ineffective. We also tried to initialize them by performing an instantaneous forward pass on the first observation, but saw no improvement.

## 5.1 MAIN RESULTS

We aim to validate our theoretical predictions that architectures with skip connections outperform those without, and that history-augmented observations will further enhance the performance of agents using skip connections according to Propositions 1 and 2.

We explore the following architectures within the parallel computation framework:

1. Default architectures: three-layer MLP or five-layer Convolutional Neural Network (CNN);
2. Augmenting observations with historical states and/or actions in the default architectures (see Appendix C for details);
3. Replacing the second last fully connected layer with an LSTM in the default architectures;
4. Adding skip connections to the default architectures;
5. Augmenting observations with historical states and/or actions in the architectures with skip connections.
6. The RLRD (Bouteiller et al., 2021) with neural execution time of one in Mujoco[3].

We tested these architectures on four Mujoco environments (HalfCheetah-v4, Walker2d-v4, Ant-v4, and Hopper-v4) and across four different neuron execution times (ranging from one to four). Additionally, we also tested these architectures on all six MinAtar environments and two toy MiniGrid environments – Random-5x5-v0 and DoorKey-5x5-v0 – where a neuron execution time of one is applied to both MinAtar and MiniGrid. A summary of results across Mujoco, MinAtar and minigrid is presented in Fig. 1b and detailed quantitative results can be found in Appendix I.

Our findings show that adding skip connections to default MLP/CNN architectures significantly enhances performance. Additionally, augmenting observations with historical states and/or actions further improves performance, aligning with Propositions 1 and 2. Fig. 3 presents more detailed results for the Mujoco environments with varying neuron execution times. It demonstrates that the agents with skip connections and state augmentation consistently match or exceed the performance of agents without skip connections and RLRD across nearly all tested environments and neuron execution times.

Moreover, as expected, in Fig. 1b we observe that neither LSTMs nor history-augmented observations offer much benefit to the default architecture without skip connections in the Mujoco or MinAtar environments. In contrast, history augmentation and LSTMs significantly improve performance in the MiniGrid environments, likely due to their underlying POMDP structure, where historical information is essential for better decision-making. We conjecture that temporal skip connection is also helpful for POMDP, as it allows agents to integrate historical data from different time steps.

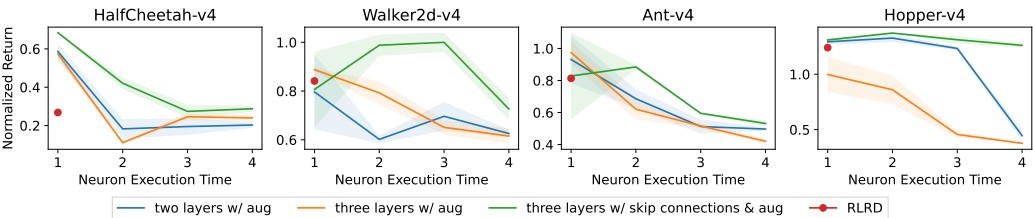

Figure 3: The performance of different agents and RLRD method on Mujoco. The agent with skip connections performs as well as, or better than, other agents in general. SAC without delay, which has a normalized performance of one, is omitted from the plots. The shaded area indicates SE across 3 seeds.

**Performance drop.** We aim to quantify the performance gap between an agent in a standard MDP without delay and our best agent in the parallel computation framework. Additionally, we are

---

[3]RLRD addresses DOMDP rather than policy-constrained DOMDP, making it not directly comparable to other choices. We use the publicly available RLRD code to obtain the results.

interested in identifying scenarios where it may be possible to close this gap between the agent in these two settings.

Fig. 3 indicates that, in many cases, there is no drop in performance when compared to the vanilla SAC without any delay. For example, this holds true for Hopper across all neuron execution times, as well as for Walker and Ant with neuron execution times of one and two.

HalfCheetah is the only Mujoco environment where a significant performance drop occurs compared to the agent without delay. To address this, we accelerated the environment making time between consecutive observation twice shorter. This adjustment resulted in a normalized performance of $0.87 \pm 0.06$ for the agent with skip connections, bringing it closer to the performance of the vanilla SAC with instantaneous actions.

In MiniGrid, the performance drop caused by parallel computations is relatively minor, whereas in MinAtar, the drop is more pronounced (refer to Appendix Tables 10 and 11). We conjecture that rendering skip connections alone insufficient to close the performance gap in MinAtar with considered neural execution time.

One potential solution could involve increasing the neural network's expressivity or reducing the neural execution time. However, if the architecture and its associated delay are fixed, the optimal solution achievable with this architecture may be strictly worse compared to an instantaneous actor, as discussed in Section 4.2.

Overall, the results show that in most environments, an agent with skip connections operating in the parallel regime can achieve performance comparable to an agent without delay, while significantly improving inference time. However, in more complex cases, skip connections alone may not be sufficient to match the performance of an agent without delay.

## 5.2 ABLATION STUDY

To identify the most effective type of skip connection, we conducted an ablation study comparing three options: projection to action, projection from observations, and a combination of projection to action with residual connections, as shown in Fig. 4. For simplicity, we refer to these sometimes as *proj-to-action*, *proj-from-obs*, and *proj-to-action & res*, respectively. Additionally we tested all possible forward skip connections between layers in Mujoco, denoting this option as *All Skips*. The results of the ablation study are summarized in Table 1. The findings help guide our selection of the default skip connection type for each environment. Based on the results, we adopt *proj-from-obs* for Mujoco environments and *proj-to-action & res* for MinAtar and MiniGrid, referring to these configurations as "skip connections" throughout the rest of the paper.

A detailed ablation study on other architectural choices, including the number of layers and augmentation strategies, is provided in AppendixC.

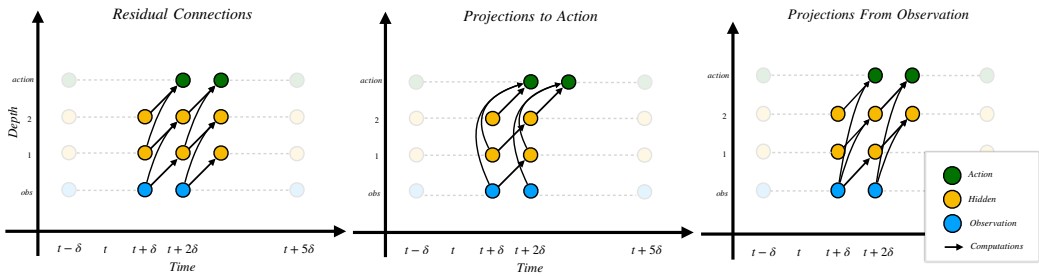

Figure 4: Illustration of different skip connections. $\delta$ represents execution time of each neuron.

**Disentangling architectural benefits of skip connections.** To disentangle the architectural benefits of temporal skip connections from their impact on reducing delays in parallel computation framework, we report the performance of a vanilla SAC pipeline, both with and without traditional skip connections and without any computational delay, in Table 2. The results show a significant

Table 1: Comparison between different skip-connection. Normalized averaged performance and standard error of agents are reported. For each task mean and SE is computed based on three seeds.

|  | **MuJoCo** | **MinAtar** | **MiniGrid** |
| --- | --- | --- | --- |
| Projections from Observation | $0.79 \pm 0.04$ | $0.45 \pm 0.05$ | $0.91 \pm .006$ |
| Projections to Action | $0.78 \pm 0.04$ | $0.46 \pm 0.04$ | $0.95 \pm .002$ |
| Projections to Action & Residual | $0.77 \pm 0.05$ | $0.52 \pm 0.06$ | $0.96 \pm .002$ |
| All Skips | $0.75 \pm 0.05$ | $-$ | $-$ |

performance improvement from traditional skip connections only in the Ant environment. Therefore, we believe that the performance gain from temporal skip connections may be due to factors beyond just reducing computational delay in only the Ant environment in parallel computation framework.

Table 2: Vanilla (without delay) SAC with and without skip connections.

|  | **Halfcheetah-v4** | **Walker2d-v4** | **Ant-v4** | **Hopper-v4** |
| --- | --- | --- | --- | --- |
| SAC | $11739 \pm 283$ | $4415 \pm 227$ | $3595 \pm 1027$ | $2672 \pm 463$ |
| SAC w/ skip connections | $11250 \pm 32$ | $4597 \pm 100$ | $5719 \pm 176$ | $2451 \pm 52$ |

## 5.3 ANALYSIS

**Distillation.** We aimed to determine whether performance limitations were due to the RL algorithm or the expressivity of our architecture. To investigate this, we used a distillation approach (employing DAgger (Ross et al., 2011)) to transfer a highly-performing vanilla SAC HalfCheetah policy (return of $11,000$) into our agent with skip connections and a neuron execution time of one. However, the distilled agent achieved a return of only $7590 \pm 93$, which was comparable to training the same architecture directly with SAC ($7892 \pm 378$). This suggests that the performance bottleneck is not algorithm-specific but rather a consequence of the reduced expressivity of the agent's architecture in capturing the true state.

**Analyzing skip connections.** We hypothesize that skip connections enable the generation of fast & effective actions, while subsequent layers refine these actions. To validate this, we removed various projection and connection pathways in a three-layer *proj-to-action* agent in Ant-v4 environment with a neuron execution time of four (Fig. 5). Specifically, we removed projections from observations, projections from the first-layer representations, and connections from the second-layer representations to the action space. The agent performed poorly without the first two projections, but still achieved some non-zero return when the connections from the last layer were removed, supporting our hypothesis.

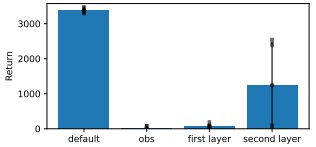

Figure 5: Removing different connections in the *proj-to-action* agent. Mean and one SD across 100 episodes are reported.

## 5.4 INFERENCE TIME SPEED-UP

We evaluated the speed-up caused by parallel computations of neurons on various hardware platforms, observing significant improvements in inference time when utilizing a GPU. Fig. 1a illustrates the percentage improvement in inference speed as the number of layers increases across different hardware configurations.

**GPU.** For GPU setting we measured performance speed-up on a single A100SXM4 GPU with 40 GB memory. The tests were conducted on a deep Multilayer Perceptron (MLP) with a batch size of one and a hidden layer size of 256 for all layers. For parallel computation on the GPU, we naively concatenated all inputs to the layers and combined all layer weights into one large sparse matrix. For agents without skip connections, this matrix has a block-diagonal form. We then used either regular

or sparse matrix multiplication to compute the output for each layer. In Fig. 1a, these approaches are labeled as GPU and GPU (sparse weights), respectively. The MLP was implemented in PyTorch, utilizing PyTorch's sparse tensor representations and sparse matrix multiplication for the GPU (sparse weights) approach.

Fig. 1a shows that the parallel computations on the GPU accelerate inference time considerably for deep neural networks. Regular matrix multiplication reached its peak performance speed-up around 30 layers, after which the speed-up started to decline; sparse matrix multiplication surpassed regular matrix multiplication at around 30 layers and continued to increase almost linearly with the number of layers achieving 350% speed-up for 100-layers MLP in our test setting.

**CPU.** We evaluated the benefits of parallelizing layers using C++ multi-threading on a CPU with 32 cores and 32 GB of RAM. Our tests showed a 6% speed-up for a 10-layer network, but gains dropped to 0.1-1% for networks with over 20 layers due to thread synchronization overhead. We used a batch size of 10,000 and hidden dimensions of 10,000 in MLP, with similar trends observed across other configurations. The limited speed-up can be attributed to the Eigen C++ library, which optimizes matrix multiplications through multi-threading, reducing the impact of further parallelization. In contrast, parallelizing naive matrix multiplications (without Eigen's optimizations) scales linearly with the number of layers, doubling for 2 layers, tripling for 3, and so on, until performance plateaus around 40 layers.

## 6 LIMITATIONS

One important assumption we make in our experiments is that we have a fixed neuron execution time ($\delta$) which is not the case in real world environments where $\delta$ can be stochastic. We propose this as a future line of work where methods can explore handling stochastic $\delta$. Additionally, we limit our experiments to at most a 5-layer neural network, as scaling vanilla RL methods to deeper architectures is non-trivial and often requires additional losses or training tricks (see Obando-Ceron et al. (2024)). Finally, we believe neuromorphic computing will benefit from our approach the most due to parallel nature of our approach. However, since neuromorphic chips are not widely available our immediate impact on the field may be limited.

## 7 CONCLUSION

Our work addresses the challenge of delays in reinforcement learning caused by parallel computations of neurons. We theoretically and experimentally show the advantages of architectures with temporal skip connections and history augmentation. These architectures demonstrate robust performance across various environments and neuron execution times. Furthermore, we demonstrate that when neuron execution time is sufficiently small, agents in the parallel regime can achieve similar performance to agents in the instantaneous regime, while significantly accelerating inference time on GPUs. This property is particularly beneficial in dynamic settings requiring rapid decision-making. However, when neuron execution times are bigger, or environments are more complex (e.g., MinAtar), the performance gap between the instantaneous and parallel regimes widens. Further research is needed to either mitigate this gap or identify cases where it may be unavoidable. Future studies could also explore asynchronous neuron computation and leverage hardware optimizations to further enhance speed-up.

## 8 ACKNOWLEDGMENT

We acknowledge the support from the Canada CIFAR AI Chair Program and from the Canada Excellence Research Chairs Program. The research was enabled in part by computational resources provided by the Digital Research Alliance of Canada and Mila Quebec AI Institute. IA thanks Nishanth Anand and Arsenii Kuznetsov for helpful discussions and comments, and Serge Zakharov for his consultation on Eigen C++.

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

## A  PPO ALGORITHM WITH PARALLEL NEURON COMPUTATION

The basic structure of the PPO algorithm is presented in Algorithm 2. To begin collecting experience, we initialize the first observation from the environment and set initial hidden activations, depicted in Fig. 2, by performing an instantaneous forward pass on the first observation. While the critic is trained online without delay, our actor is trained within the in-parallel computation framework by unrolling it on recent sub-trajectories stored in the buffer (with hidden activation reset with instantaneous forward pass), allowing all weights to be available for backpropagation.

Typically, in PPO, the critic and actor share a common backbone. However, to enable online training of the critic without delay, we employ separate neural networks for the critic and actor.

---

**Algorithm 2** PPO with parallel neuron computation.

---

1: Init an actor and a critic with random parameters.
2: Set initial state to be $s_0, h_0^0, ..., h_0^N$ , where $h_0^j$ is activations for layer $j$ at a time step $0$.
3: Wrap the environment with sticky actions or repeating observations wrapper if needed based on neural execution time.
4: **for** $t \in 0, \ldots, L$ **do**
5:     $a_t, h_{t+1}^0, ..., h_{t+1}^N = Actor(s_t, h_t^0, ..., h_t^N)$ (Query current policy for the next action and next Actor's hidden activations given current observation and hidden activations)
6:     Take the action $a_t$ and receive $\{r_t, s_{t+1}\}$ from the environment and put $\{s_t, a_t, r_t\}$ to the buffer.
7:     **if** buffer is full **then**
8:         Compute gae return on the buffer.
9:         **for** $t \in 0, \ldots, $ n_epochs **do**
10:             Init $h_0^0, ..., h_0^N$ and simulate the actor dynamic forward and get critic output without delay on the collected buffer.
11:             Compute the PPO loss
12:             Update the actor and the critic (via back-propagation through time if needed)
13:         **end for**
14:         Empty the buffer
15:     **end if**
16: **end for**

---

## B  SUPPLEMENTARY EXPERIMENTAL RESULTS

**Atari games.**  We present preliminary results on a small subset of Atari environment in Table 3. We use the same architectures and hyper-parameters as we used for MiniGrid experiments. As standard choice in Atari we augment the state with 4 past observations, grayscaled observations, for all actors and bin reward to be $+1, 0, -1$ by its sign, and repeat each action four times for all agents.

Table 3: Subset of Atari games average returns after training on 1 mln observations. The results are averaged across three seeds, mean and standard deviation are reported. PPO denotes vanilla PPO without inference delay. CNN and CNN with skip connections denote architecture executed withing parallel computation framework with neural execution time of one.

|  | PPO | CNN w/ aug | CNN w/ skip & aug |
|---|---|---|---|
| **Boxing-v5** | $21.4 \pm 4.4$ | $2.8 \pm 1.4$ | $2.6 \pm 3.6$ |
| **Breakout-v5** | $15.6 \pm 7.6$ | $6.3 \pm 0.7$ | $6.9 \pm 5.5$ |
| **BattleZone-v5** | $3683 \pm 375$ | $5336 \pm 1366$ | $4943 \pm 984$ |
| **SpaceInvaders-v5** | $386 \pm 27$ | $412 \pm 8$ | $456 \pm 31$ |
| **Assault-v5** | $887 \pm 186$ | $660 \pm 9$ | $712 \pm 37$ |
| **Bowling-v5** | $37.2 \pm 2.4$ | $37.9 \pm 4.9$ | $40.9 \pm 9.6$ |
| **Freeway-v5** | $22.18 \pm 0.34$ | $22.18 \pm 0.34$ | $22.18 \pm 0.34$ |

**Stochastic environments.**   We also conducted experiments in stochastic environments by introducing "sticky actions" in MinAtar, where the agent's last action was repeated with a probability of 0.25 (Table 4). This modification led to a decline in performance across all agents; however, the relative trends remained consistent.

Table 4: Results after training on 10 million samples in MinAtar games with a sticky action probability of 0.25. The mean and standard error across three seeds are reported. The neuron execution time is 1. PPO refers to the standard implementation of PPO without inference delay.

|  | Breakout-v0 | Seaquest-v0 | Freeway-v0 | Asterix-v0 | SpaceInv-v0 |
|---|---|---|---|---|---|
| PPO | $8.29 \pm 1.16$ | $21.48 \pm 8.55$ | $60.15 \pm 1.53$ | $25.38 \pm 2.40$ | $91.38 \pm 12.33$ |
| CNN w/ aug | $3.38 \pm 0.32$ | $2.72 \pm 1.15$ | $27.59 \pm 2.49$ | $4.25 \pm 1.86$ | $25.08 \pm 0.43$ |
| CNN w/ skip & aug | $6.29 \pm 0.18$ | $5.38 \pm 0.83$ | $53.51 \pm 0.84$ | $9.94 \pm 2.03$ | $40.47 \pm 1.04$ |

**Sequential baseline.**   Throughout the paper, we use an agent with pipelining (parallel computation of layers) but without skip connections as our simplest baseline. In Table 5, we also present a baseline for an agent that computes layers sequentially (see the leftmost graph in Fig. 2). To construct this baseline, we needed to estimate how much slower an agent would be without parallel computations. To highlight the potential benefits of pipelining, we assumed an ideal speed-up scenario for parallelization, representing the performance gain at its theoretical limit. Specifically, for a three-layer neural network, we assumed a threefold slowdown when abandoning parallelization.

It is important to note that Table 5 reinterprets the information available in Fig. 3. We report the sequential agent for MinAtar in Table 10.

Table 5: Mujoco average normalized returns after 1mln states of training for the four selected environments. Neural execution time is one.

|  | Halfcheetah-v4 | Walker2d-v4 | Ant-v4 | Hopper-v4 |
|---|---|---|---|---|
| sequential three layers w/ aug | 0.246 | 0.651 | 0.516 | 0.456 |
| three layers w/ aug | 0.574 | 0.888 | 0.974 | 0.998 |
| three layers w/ skip & aug | 0.685 | 0.807 | 0.828 | 1.309 |

## C   SUPPLEMENTARY ABLATION RESULTS

**Varying number of layers.**   We are interested in how the number of layers impacts the performance of the agents. Our hypothesis is that performance will be highly sensitive to the number of layers in a default MLP, as it directly influences the amount of delay. In contrast, we expect the sensitivity to be lower for MLPs with skip connections. Additionally, we aim to investigate whether an architecture with a well-tuned number of layers, but without skip connections, could outperform one that includes skip connections.

Figure 6 shows that the optimal number of layers without skip connections for Mujoco environments in the parallel pipeline is two. However, this configuration does not outperform the MLP with skip connections. In fact, the results are even stronger: as shown in Figure 3, the three-layer MLP with skip connections consistently outperforms both two- and three-layer MLPs without skip connections across nearly all environments and neural execution times.

Additionally, we varied the number of layers in the augmented agent with skip connections (having a neuron execution time of four) across three Mujoco environments, as illustrated in Figure 10. We found that increasing the number of layers from two to three improved performance in all environments, a trend also supported by the last two bars in Figure 6. However, beyond three layers, the performance trends diverged and stopped being statistically significant, leading us to adopt three layers as the default choice for skip-connected MLPs. Notably, performance does not significantly

drop when exceeding three layers, suggesting that the architecture with skip connections adapts the effective number of layers to manage the delay.

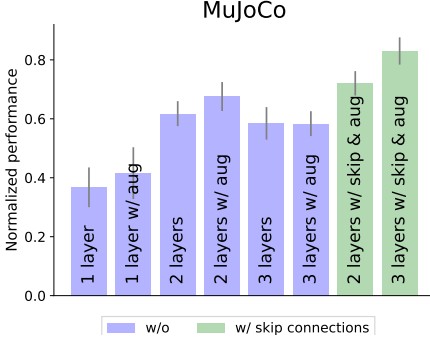

Figure 6: Ablating number of layers in Mujoco.

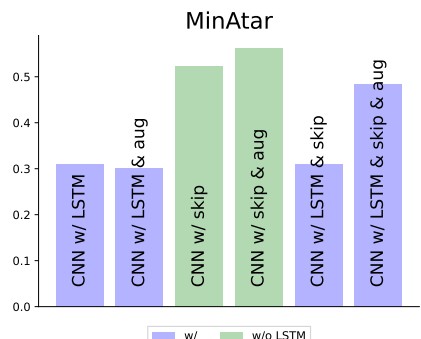

Figure 7: Recurrence with skip connections.

**Combining recurrent and skip connections.**   One way to make agents more expressive without increasing delay is to add recurrent connections. We investigated whether this result in better performance. We experimented with combining recurrent (LSTM) and skip connections. However, this combination degraded performance on MinAtar (see Fig. 7) or failed to provide notable improvements on MiniGrid (see Table 11). We believe that combination of LSTM and skip connection may require additional tuning of hyperparameters.

**Action repetition.**   We included SAC-repeat-2 and SAC-repeat-3, which are variants of the vanilla instantaneous SAC algorithm with action repetition, as part of our ablation study (Table 6). In these versions, the same action is repeated in the environment two or three times, respectively. This can improve overall performance in MuJoCo environments by simplifying the credit assignment problem. Our findings show that action repetition significantly enhance performance on the Ant and Hopper tasks with two repetitions, and on Hopper with three repetitions. We believe this makes action repetition particularly responsible for the good results in a parallel computations setting, when the neural execution time is set to two for Ant and Hopper, and three for Hopper.

Table 6: Average returns after 1mln states of training in the four selected environments for SAC and SAC with sticky actions. The results are averaged across 3 seeds. Mean and standard error are reported.

|  | **Halfcheetah-v4** | **Walker2d-v4** | **Ant-v4** | **Hopper-v4** |
|---|---|---|---|---|
| SAC | $\mathbf{11739 \pm 283}$ | $4415 \pm 227$ | $3595 \pm 1027$ | $2672 \pm 463$ |
| SAC-repeat-2 | $8626 \pm 523$ | $\mathbf{4670 \pm 221}$ | $\mathbf{4102 \pm 1228}$ | $\mathbf{3520 \pm 140}$ |
| SAC-repeat-3 | $8168 \pm 618$ | $3763 \pm 582$ | $2625 \pm 796$ | $3517 \pm 94$ |

**Ablating observation augmentation strategies.**   Following common practices, we augment observations with four past frames in MiniGrid to account for its original partial observability, two recent actions in Mujoco, and one recent action in MinAtar, based on Proposition 2 and the ablation study results presented here.

We conducted an ablation study to investigate different observation augmentation strategies by varying the number of recent available actions included in state augmentation for Mujoco, ranging number of actions from one to three. Fig. 8 presents the results for two architectures: three-layer MLP and three-layer MLP with skip connections. While there is no significant difference in performance for the standard three-layer MLP, the MLP with skip connections shows a slight performance improvement when augmenting the state with the two most recent available actions. Based on these findings, we use state augmentation with two actions as the default choice for Mujoco environments.

Similarly, Fig. 9 shows the results for MinAtar. We experimented with three augmentation strategies: using the four most recent available states, adding the most recent available action to the hidden representations of the last two fully connected layers, and a combination of both. Interestingly, all these strategies resulted in approximately the same performance improvement for the CNN with skip connections. Therefore, we chose the simpler and theoretically supported approach of augmenting with the last available action as the default strategy in MinAtar environments.

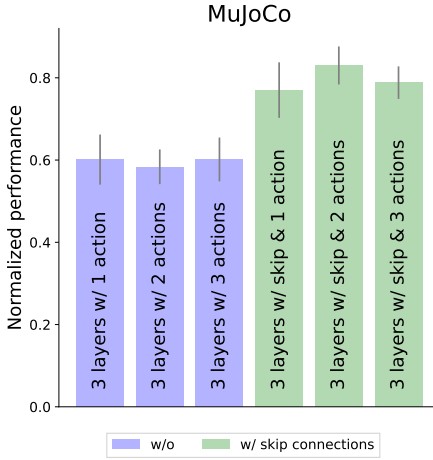
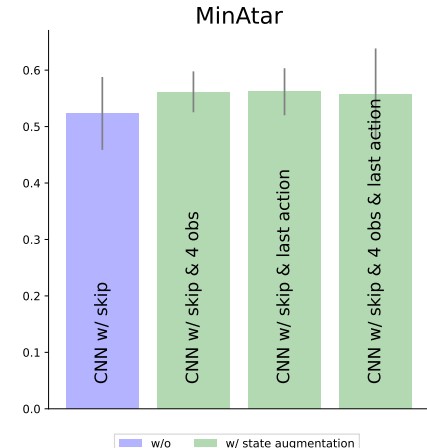

Figure 8: Ablating augmentation choices in Mujoco.

Figure 9: Ablating augmentation choices in MinAtar.

# D ADDITIONAL ANALYSIS

**Noisy computations.** In our simulations, while we model parallel neuron computations during both inference and training, the processes were globally synchronized, meaning that all neurons completed and initiated new computations simultaneously. As a step towards introducing asynchronous neuron computations, we tested a noisier version of parallel computation by applying dropout in every hidden layer during the training and inference stages in our agent with skip connections. In Fig. 11 one can see that the agent is quite robust to a large amount of dropout, and the performance starts to deteriorate if dropout probability becomes more than 40%. The motivation behind this approach comes from the fact that when each neuron updates asynchronously, we can track the time elapsed since the last update and if this time exceeds a predefined threshold, we can zero out the activation, mimicking the effect of dropout to some extent.

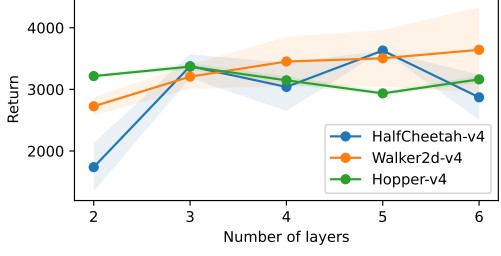
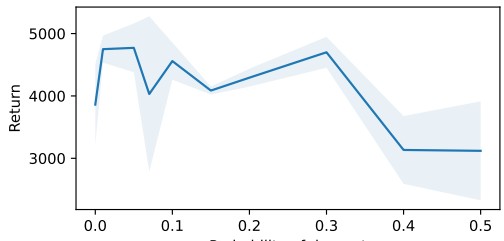

Figure 10: Varying number of layers in the agent with skip connections having neuron execution time of four. The shaded area indicates the standard error.

Figure 11: Average return with SE across 3 seeds vs different amounts of dropout for the agent with skip connections in HalfCheetah with neuron execution time of two.

**Qualitative analysis.** Trajectories rollouts of the CNN agent with skip connections and the CNN agent (without skip connections and with history augmentation) is presented in Fig. 12 for MiniGrid-

DoorKey-5x5-v0. The objective in the game is to find the key, toggle the door and reach the destination. The trajectories show the agent with skip connections demonstrates less "roaming around" behaviour compared to the agent without skip connections. inal location while the agent without skip connections is fairly indecisive. Notably, it took the agent without skip connections 2x more steps on an average to reach the goal compared to the one with skip connections. We present multiple trajectory samples in Appendix J.

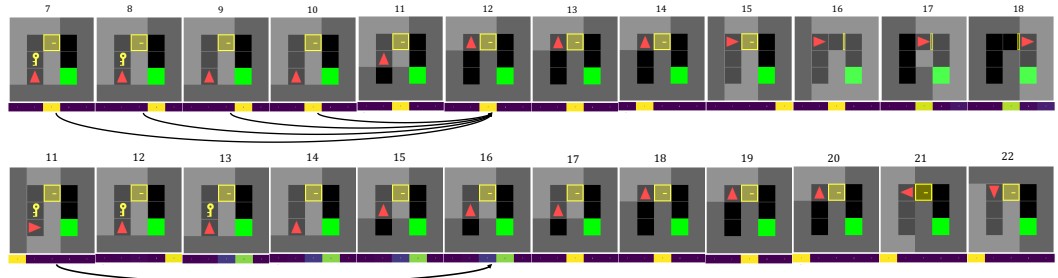
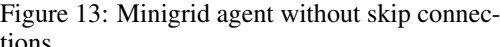

Figure 12: Behaviour of agents with (top row) and without (bottom row) skip connections on MiniGrid-DoorKey-5x5. For comparison, we pick a sub-trajectory from the full episode for each agent. The arrows in each figure indicate the observations that influence the decision-making process. For the sake of brevity, we have shown only one set of temporal connection in both the cases. However, they exist throughout. The heatmap below each figure denotes the action probabilities, the actions in this case are (in the same sequence in the heatmap): `l:turn left,r:turn right,f:move forward, p:pickup an object, t:  toggle an object`. The agent with skip connections show less "wandering" behaviour. For instance, the agent with no skip connections reaches the door and continues to take random actions while the agent with skip connections toggles the door much earlier. Interestingly, the agent without skip connection is quite confident in its decisions. We hypothesize that because an agent's own policy makes the environment appear non-stationary, high confidence may help it cope with this.

# E  IMPLEMENTATION DETAILS AND HYPERPARAMETERS USED IN EXPERIMENTS

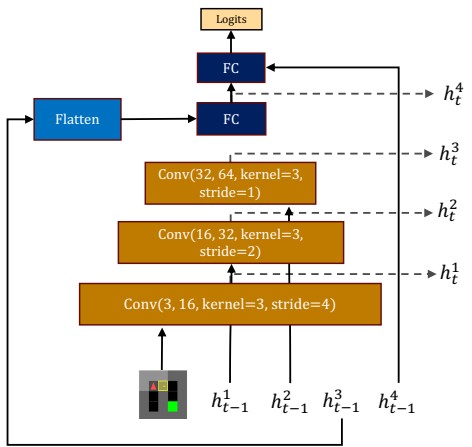

Figure 13: Minigrid agent without skip connections

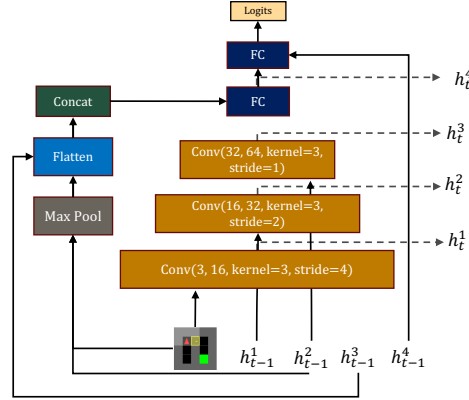

Figure 14: Minigrid agent with skip connections. Residual connections are emitted from the figure for simplicity.

We use ReLu activation function. Our SAC actor employees three-layer MLP if not stated otherwise with hidden dimensions of 256.

Our PPO actor employs a 3-layer Convolutional Neural Network (CNN) followed by two fully connected layer with hidden dimension of 512. All CNN layers have a kernel size of 3 and $C = \{32, 64, 64\}$ channels, maintaining the same resolution throughout the CNN for MinAtar and using strides $\{4, 2, 1\}$ for MiniGrid. The feature volume is then flattened and fed into the fully connected layer for action prediction. For architectures with skip connections, the feature volumes from previous layers are maxpooled, concatenated and then flattened and subsequently fed to the fully connected layers. Our Q-network shares the same architecture as the actor.

Notably, when working with networks incorporating skip connections, we observed a performance drop when attempting to combine all convolutional features by flattening and concatenating them into a single feature volume. To address this, we experimented with various methods for feature combination and found that **max-pooling** all features to a fixed size before flattening and concatenating yielded the best results.

We present the architecture with skip connections used for the MiniGrid experiments in Figure 14; a similar architecture was also employed for MinAtar. Specifically, given an input $x_t$ at time step $t$ and a set of hidden activations $(h_{t-1}^1, h_{t-1}^2, \dots)$, if these are convolutional features, they are max-pooled using the formula:

$$size = \frac{current\_spatial\_size}{last\_spatial\_size}$$

Here, *current_spatial_size* refers to the spatial size of the current convolutional feature, and *last_spatial_size* refers to the spatial size of the final convolutional feature in the network (the third convolutional block in Fig. 14). After max-pooling, the features are flattened, concatenated, and passed through linear layers for further processing.

The hyperparameters used in the main experiments on SAC Mujoco and PPO MinAtar/MiniGrid can be found in Table 7. For training LSTM in Mujoco, we used a learning rate of 1e-4 instead of the default value specified in the table, as we observed a slight improvement in performance with this adjustment.

## F  DEFINING REGRETS

We define regret with respect to cumulative undiscounted rewards:

$$G^\pi(t, \pi') = \mathbb{E}\left[ \sum_{i < t,\, i \in D(\pi')} r_i \,\middle|\, \pi \right]$$

where $i \in D(\pi')$ indicates the time steps where the policy $\pi'$ is used. The expectation is over trajectories generated by following policy $\pi$, but rewards are accumulated only at times $i \in D(\pi')$.

We denote $\pi^*$ as an optimal policy in original MDP and $\pi^*_{\text{parallel}}$ as an optimal policy in parallel computation framework. Then delay regret, $\Delta_{\text{delay}}(t)$, and inaction regret, $\Delta_{\text{inaction}}(t)$, are defined as:

$$\Delta_{\text{delay}}(t) := G^{\pi^*}(t, \pi^*_{\text{parallel}}) - G^{\pi^*_{\text{parallel}}}(t, \pi^*_{\text{parallel}}) \tag{3}$$

$$\Delta_{\text{inaction}}(t) := G^{\pi^*}(t, \beta) - G^{\beta}(t, \beta) \tag{4}$$

where $\beta$ is default policy we employ during inaction period.

Table 7: Hyperparameters used in experiments.

| Parameter | Value |
|---|---|
| **SAC Mujoco** | |
| Discount rate $\gamma$ | 0.99 |
| Policy frequency | 2 |
| Target network frequency | 1 |
| Target smoothing coefficient | 0.005 |
| Policy learning rate | 3e-4 |
| Q-function learning rate | 1e-3 |
| Optimizer | Adam |
| Adam beta | (0.9, 0.999) |
| Adam epsilon | 1e-8 |
| Replay buffer size | 1,000,000 |
| Batch size | 256 |
| Learning starts | 10,000 |
| Entropy regularization | Auto-tuned |
| Target entropy scale | 1 |
| | |
| **PPO MinAtar and MiniGrid** | |
| Discount rate $\gamma$ | 0.99 |
| Lambda for general advantage estimation | 0.95 |
| Entropy coefficient | 0.01 |
| Value function coefficient | 0.5 |
| Normalize advantages | True |
| Number of steps to unroll a policy | 32 |
| Number of environments | 32 |
| Update epochs | 4 |
| Learning rate | 2.5e-4 |
| Anneal lr | True |
| Optimizer | Adam |
| Adam beta | (0.9, 0.999) |
| Adam epsilon | 1e-5 |
| Maximum gradient norm for clipping | 0.5 |

# G    DISCUSSION OF REALTIME SETTING AND INACTION REGRET

In realtime RL settings agents and environments interact asynchronously at their own pace. As discussed by Travnik et al. (2018), this setting is more realistic of real world deployment than the typical sequential interaction paradigm of RL where the agent and environment are both assumed to wait for each other in a turn-based manner. Sources of regret as a function of time were recently analyzed for this setting by Riemer et al. (2024) where it was concluded that the total accumulated realtime regret $\Delta_{\text{realtime}}$ can be decomposed into three different sources such that $\Delta_{\text{realtime}} = \Delta_{\text{inaction}} + \Delta_{\text{delay}} + \Delta_{\text{learn}}$. Here $\Delta_{\text{learn}}$ is the typical kind of regret analyzed in RL (Kearns & Singh, 2002) that arises from the need for the algorithm to explore and learn from its environment. This kind of regret is present even in standard turn-based environments that can pause. However, $\Delta_{\text{inaction}}$ and $\Delta_{\text{delay}}$ are new notions of regret specific to the realtime setting. $\Delta_{\text{inaction}}$ is regret incurred when an agent does not act frequently enough in the environment. Meanwhile, $\Delta_{\text{delay}}$ is the regret incurred because actions are produced based on delayed observations.

**Proposition 3** *(Inaction Regret): In parallel computation framework for any $N$ layer neural network constrained such that $\delta \leq 1$, inaction regret is zero. Otherwise, if $\delta \geq 1$ the regret from inaction is independent of $N$ and bounded by:*

$$\Delta_{inaction}(t) \in \Theta(t(\delta - 1)) \tag{5}$$

This proposition follows from the fact that inference time speedups within the parallel computation framework allow a network of length $N$ to achieved an $N$ times increased action throughput. This result then follows when considering the worst case environment from (Riemer et al., 2024) Theorem 1 where the default behavior is always sub-optimal when the agent does not act with its own policy. This bound with the parallel computation framework is significantly better for large networks than the bound of $\Delta_{\text{inaction}}(t) \in \Theta(t(N\delta - 1)$ with a single standard sequential layer inference process.

# H    TRAINING & HARDWARE

We used A100SXM4 GPU for training all our methods. We use the same GPU for testing our methods as well. It took us approximately 2 hours per seed to train a MinAtar experiment for 10 million steps, 4 hours per seed – MiniGrid experiment, and it took 7 hours per seed to train one MuJoCo experiment for 1 million steps.

# I   DETAILED UNNORMALIZED RESULTS

Detailed unnormalized results for all main and ablation study experiments are provided in Tables 8, 10 and 11 for Mujoco, MinAtar, and MiniGrid, respectively.

For MinAtar, we additionally include results for a neural execution time of 0.4 and 2. For MiniGrid, we also provide results with a neural execution time of 4. The neural execution time of $0.4$ for the MinAtar five-layer CNN agent is achieved by treating the computation of 2.5 layers as a new basic block within the parallel computation pipeline.

Table 8: Mujoco average returns after 1mln states of training for the four selected environments. The results are averaged across ten seeds for two layers, three layers and w/ projections from observation agents without augmentations, across 5 seeds for RLRD and the rest of results use 3 seeds. Mean and standard error are reported. We take mean action as a policy during evaluation stage as we notice it may significantly boost the performance of the delayed actor. RLRD (Bouteiller et al., 2021) is a baseline that addresses DOMDP rather than a parallel computations.

| | Halfcheetah-v4 | Walker2d-v4 | Ant-v4 | Hopper-v4 |
|---|---|---|---|---|
| SAC | $11739 \pm 283$ | $4415 \pm 227$ | $3595 \pm 1027$ | $2672 \pm 463$ |
| | neuron execution time of 1 | | | |
| RLRD for delay of 1 | $3147 \pm 1044$ | $3714 \pm 547$ | $2924 \pm 568$ | $3314 \pm 157$ |
| one layer | $5086 \pm 662$ | $1209 \pm 652$ | $1043 \pm 526$ | $759 \pm 65$ |
| two layers | $6660 \pm 360$ | $4271 \pm 164$ | $1938 \pm 274$ | $3001 \pm 293$ |
| three layers | $7814 \pm 130$ | $4459 \pm 176$ | $2792 \pm 620$ | $3115 \pm 173$ |
| LSTM | $7096 \pm 138$ | $3764 \pm 410$ | $2847 \pm 587$ | $2462 \pm 162$ |
| three layers w/ proj-to-action & res | $8295 \pm 522$ | $4567 \pm 123$ | $4425 \pm 424$ | $3019 \pm 432$ |
| three layers w/ proj-to-action | $7690 \pm 480$ | $4048 \pm 681$ | $4599 \pm 111$ | $2871 \pm 552$ |
| three layers w/ proj-from-obs | $7892 \pm 379$ | $4497 \pm 140$ | $3728 \pm 355$ | $3187 \pm 195$ |
| three layers w/ all skips | $8102 \pm 285$ | $4934 \pm 57$ | $4270 \pm 410$ | $3381 \pm 114$ |
| two layers w/ aug | $6881 \pm 467$ | $3516 \pm 681$ | $3347 \pm 532$ | $3454 \pm 76$ |
| three layers w/ aug | $6735 \pm 387$ | $3920 \pm 33$ | $3502 \pm 252$ | $2666 \pm 422$ |
| LSTM w/ aug | $7980 \pm 168$ | $3289 \pm 207$ | $2167 \pm 335$ | $2948 \pm 317$ |
| two layers w/ proj-from-obs & aug | $7082 \pm 266$ | $4596 \pm 376$ | $2969 \pm 48$ | $3389 \pm 148$ |
| three layers w/ proj-from-obs & aug | $8037 \pm 201$ | $3561 \pm 682$ | $2976 \pm 972$ | $3499 \pm 30$ |
| three layers w/ all skips & aug | $8165 \pm 196$ | $4490 \pm 174$ | $4729 \pm 267$ | $3253 \pm 229$ |
| | neuron execution time of 2 | | | |
| one layer | $1846 \pm 440$ | $1535 \pm 329$ | $1783 \pm 243$ | $1980 \pm 439$ |
| two layers | $3173 \pm 399$ | $3791 \pm 368$ | $2061 \pm 108$ | $3317 \pm 31$ |
| three layers | $3027 \pm 329$ | $3277 \pm 180$ | $1780 \pm 276$ | $2538 \pm 281$ |
| LSTM | $2413 \pm 363$ | $3084 \pm 92$ | $2078 \pm 92$ | $2764 \pm 150$ |
| three layers w/ proj-to-action & res | $3330 \pm 751$ | $4226 \pm 258$ | $2049 \pm 101$ | $3531 \pm 61$ |
| three layers w/ proj-to-action | $4729 \pm 145$ | $3197 \pm 396$ | $2685 \pm 19$ | $3597 \pm 14$ |
| three layers w/ proj-from-obs | $4715 \pm 392$ | $4137 \pm 291$ | $2669 \pm 224$ | $3569 \pm 63$ |
| three layers w/ all skips | $1682 \pm 80$ | $3068 \pm 415$ | $1472 \pm 900$ | $3607 \pm 17$ |
| two layers w/ aug | $2142 \pm 598$ | $2656 \pm 69$ | $2467 \pm 214$ | $3544 \pm 94$ |
| three layers w/ aug | $1298 \pm 67$ | $3499 \pm 209$ | $2231 \pm 224$ | $2297 \pm 345$ |
| LSTM w/ aug | $3027 \pm 636$ | $2724 \pm 217$ | $1459 \pm 278$ | $1794 \pm 467$ |
| two layers w/ proj-from-obs & aug | $4156 \pm 116$ | $2774 \pm 114$ | $2255 \pm 145$ | $3628 \pm 30$ |
| three layers w/ proj-from-obs & aug | $4937 \pm 351$ | $4362 \pm 194$ | $3180 \pm 51$ | $3665 \pm 9$ |
| three layers w/ all skips & aug | $3649 \pm 416$ | $4419 \pm 152$ | $3172 \pm 168$ | $3580 \pm 22$ |

Table 9: Continuation of the Table 8.

|  | Halfcheetah-v4 | Walker2d-v4 | Ant-v4 | Hopper-v4 |
|---|---|---|---|---|
| neuron execution time of 3 | | | | |
| one layer | $1293 \pm 481$ | $2157 \pm 189$ | $1597 \pm 110$ | $1150 \pm 347$ |
| two layers | $2299 \pm 350$ | $3037 \pm 180$ | $1884 \pm 125$ | $2098 \pm 97$ |
| three layers | $3145 \pm 158$ | $3054 \pm 106$ | $1865 \pm 79$ | $1171 \pm 79$ |
| LSTM | $2816 \pm 254$ | $2843 \pm 324$ | $987 \pm 690$ | $1144 \pm 293$ |
| three layers w/ proj-to-action & res | $2897 \pm 679$ | $2711 \pm 245$ | $1205 \pm 380$ | $3647 \pm 15$ |
| three layers w/ proj-to-action | $2886 \pm 147$ | $2391 \pm 6$ | $2160 \pm 66$ | $3633 \pm 33$ |
| three layers w/ proj-from-obs | $3224 \pm 424$ | $3043 \pm 182$ | $2097 \pm 84$ | $3505 \pm 43$ |
| three layers w/ all skips | $2745 \pm 733$ | $2583 \pm 8$ | $2182 \pm 208$ | $3290 \pm 153$ |
| two layers w/ aug | $2286 \pm 511$ | $3074 \pm 261$ | $1842 \pm 162$ | $3294 \pm 42$ |
| three layers w/ aug | $2886 \pm 204$ | $2874 \pm 133$ | $1856 \pm 59$ | $1218 \pm 80$ |
| LSTM w/ aug | $2212 \pm 509$ | $2700 \pm 299$ | $1085 \pm 261$ | $1291 \pm 35$ |
| two layers w/ proj-from-obs & aug | $1729 \pm 616$ | $3006 \pm 133$ | $566 \pm 699$ | $3603 \pm 24$ |
| three layers w/ proj-from-obs & aug | $3214 \pm 417$ | $4415 \pm 174$ | $2139 \pm 14$ | $3504 \pm 90$ |
| three layers w/ all skips & aug | $3459 \pm 440$ | $3237 \pm 179$ | $1927 \pm 191$ | $3647 \pm 7$ |
| neuron execution time of 4 | | | | |
| one layer | $1284 \pm 251$ | $1975 \pm 159$ | $1132 \pm 561$ | $1372 \pm 151$ |
| two layers | $1681 \pm 281$ | $2355 \pm 291$ | $1427 \pm 467$ | $1263 \pm 91$ |
| three layers | $2421 \pm 133$ | $2532 \pm 109$ | $735 \pm 756$ | $1041 \pm 21$ |
| LSTM | $2496 \pm 303$ | $2353 \pm 58$ | $617 \pm 747$ | $724 \pm 176$ |
| three layers w/ proj-to-action & res | $1748 \pm 414$ | $2886 \pm 186$ | $1909 \pm 178$ | $3472 \pm 78$ |
| three layers w/ proj-to-action | $2330 \pm 594$ | $3079 \pm 319$ | $1642 \pm 29$ | $3310 \pm 35$ |
| three layers w/ proj-from-obs | $2674 \pm 288$ | $2898 \pm 134$ | $1959 \pm 78$ | $2990 \pm 209$ |
| three layers w/ all skips | $1733 \pm 408$ | $3031 \pm 86$ | $1898 \pm 159$ | $3016 \pm 128$ |
| two layers w/ aug | $2378 \pm 177$ | $2762 \pm 69$ | $1785 \pm 36$ | $1190 \pm 136$ |
| three layers w/ aug | $2813 \pm 206$ | $2716 \pm 139$ | $1512 \pm 56$ | $1005 \pm 11$ |
| LSTM w/ aug | $2206 \pm 168$ | $2792 \pm 212$ | $1702 \pm 62$ | $892 \pm 55$ |
| two layers w/ proj-from-obs & aug | $1738 \pm 393$ | $2726 \pm 147$ | $1862 \pm 126$ | $3215 \pm 26$ |
| three layers w/ proj-from-obs & aug | $3375 \pm 193$ | $3206 \pm 197$ | $1911 \pm 66$ | $3369 \pm 53$ |
| three layers w/ all skips & aug | $2879 \pm 437$ | $3074 \pm 134$ | $1906 \pm 53$ | $2711 \pm 293$ |

## J FULL TRAJECTORY ROLLOUTS FOR MINIGRID

We present a few full trajectories of agents with and without skip-connections on Fig. J.

Table 10: Full results for PPO MinAtar. Average returns after training on 10 million samples on MinAtar games. Results are averaged across three seeds and standard error is reported. Sequential CNN w/ aug represents an agent that computes layers sequentially and we assume the speed of the inference of this five-layers neural network is five times slower.

| | Breakout-v0 | Seaquest-v0 | Freeway-v0 | Asterix-v0 | SpaceInv-v0 |
|---|---|---|---|---|---|
| PPO | $20.81 \pm 0.15$ | $25.94 \pm 14.52$ | $64.68 \pm 0.93$ | $42.01 \pm 0.52$ | $297.49 \pm 69.86$ |
| | neuron execution time of 0.4 | | | | |
| CNN | $16.88 \pm 0.81$ | $25.75 \pm 3.65$ | $57.28 \pm 0.51$ | $9.89 \pm 3.82$ | $78.55 \pm 2.24$ |
| | neuron execution time of 1 | | | | |
| Sequential CNN w/ aug | $0.79 \pm 0.06$ | $1.51 \pm 0.32$ | $28.68 \pm 0.18$ | $2.24 \pm 0.20$ | $14.29 \pm 0.93$ |
| CNN | $7.92 \pm 0.84$ | $8.41 \pm 0.05$ | $28.865 \pm 4.31$ | $7.81 \pm 2.27$ | $35.39 \pm 1.52$ |
| CNN w/ aug | $6.67 \pm 0.27$ | $7.10 \pm 1.75$ | $28.31 \pm 2.61$ | $10.41 \pm 0.50$ | $41.95 \pm 1.51$ |
| LSTM | $6.49 \pm 0.43$ | $5.63 \pm 1.01$ | $28.85 \pm 0.46$ | $7.35 \pm 1.34$ | $34.31 \pm 1.26$ |
| LSTM w/ aug | $4.20 \pm 1.31$ | $4.45 \pm 1.48$ | $29.50 \pm 0.77$ | $7.68 \pm 2.06$ | $37.26 \pm 0.37$ |
| CNN w/ skip | $14.46 \pm 1.81$ | $15.59 \pm 4.26$ | $52.11 \pm 2.43$ | $11.74 \pm 1.20$ | $69.80 \pm 1.42$ |
| CNN w/ skip & aug | $16.96 \pm 0.58$ | $16.01 \pm 3.34$ | $50.79 \pm 2.06$ | $13.24 \pm 1.09$ | $73.58 \pm 3.72$ |
| CNN w/ skip & aug & lstm | $11.69 \pm 0.56$ | $4.16 \pm 2.61$ | $16.80 \pm 0.72$ | $1.23 \pm 0.54$ | $52.37 \pm 1.41$ |
| | neuron execution time of 2 | | | | |
| CNN | $2.53 \pm 0.18$ | $3.22 \pm 1.10$ | $31.29 \pm 0.30$ | $8.61 \pm 0.55$ | $29.35 \pm 2.23$ |
| LSTM | $2.68 \pm 0.04$ | $4.14 \pm 1.50$ | $29.74 \pm 1.22$ | $5.25 \pm 1.96$ | $34.84 \pm 2.20$ |
| CNN w/ skip | $5.41 \pm 0.27$ | $9.65 \pm 1.09$ | $40.84 \pm 1.70$ | $9.28 \pm 1.01$ | $53.50 \pm 3.67$ |

Table 11: MiniGrid average returns after training on 10 mln states for the two toy environments. The agent receives a reward of one only when reaching the target location. The results are averaged across three seeds, mean and standard error are reported. PPO denotes vanilla PPO without inference delay.

| | Empty-Random-5x5-v0 | DoorKey-5x5-v0 |
|---|---|---|
| PPO | $0.963 \pm 0.0013$ | $0.961 \pm 0.0015$ |
| | neuron execution time of 1 | |
| CNN | $0.812 \pm 0.0137$ | $0.613 \pm 0.0053$ |
| CNN w/ aug | $0.894 \pm 0.0036$ | $0.859 \pm 0.0064$ |
| LSTM | $0.922 \pm 0.0038$ | $0.904 \pm 0.0074$ |
| LSTM w/ aug | $0.855 \pm 0.0343$ | $0.895 \pm 0.0134$ |
| CNN w/ skip | $0.924 \pm 0.0025$ | $0.930 \pm 0.0020$ |
| CNN w/ skip & aug | $0.932 \pm 0.0031$ | $0.922 \pm 0.0047$ |
| CNN w/ skip & lstm | $0.926 \pm 0.0055$ | $0.920 \pm 0.0097$ |
| CNN w/ skip & aug & lstm | $0.933 \pm 0.0024$ | $0.932 \pm 0.0018$ |
| | neuron execution time of 4 | |
| CNN | $0.810 \pm 0.0043$ | $0.607 \pm 0.0118$ |
| CNN w/ aug | $0.894 \pm 0.0075$ | $0.872 \pm 0.0019$ |
| LSTM | $0.919 \pm 0.0027$ | $0.899 \pm 0.0085$ |
| LSTM w/ aug | $0.916 \pm 0.0059$ | $0.788 \pm 0.1088$ |
| CNN w/ skip | $0.923 \pm 0.0044$ | $0.933 \pm 0.0010$ |
| CNN w/ skip & aug | $0.921 \pm 0.0025$ | $0.919 \pm 0.0026$ |
| CNN w/ skip & lstm | $0.933 \pm 0.0011$ | $0.567 \pm 0.2333$ |
| CNN w/ skip & aug & lstm | $0.923 \pm 0.0074$ | $0.927 \pm 0.0033$ |

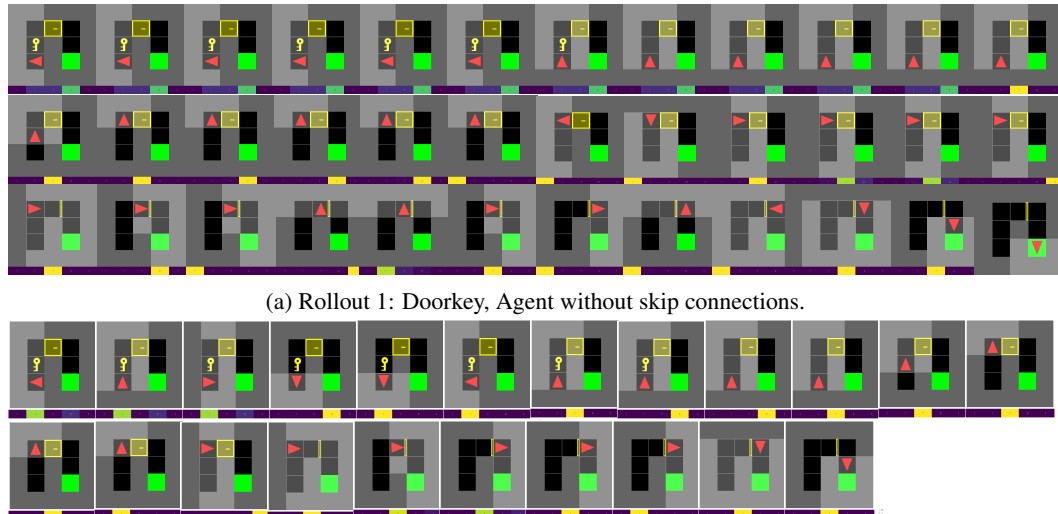

(a) Rollout 1: Doorkey, Agent without skip connections.

(b) Rollout 1: Doorkey, Agent with skip connections.

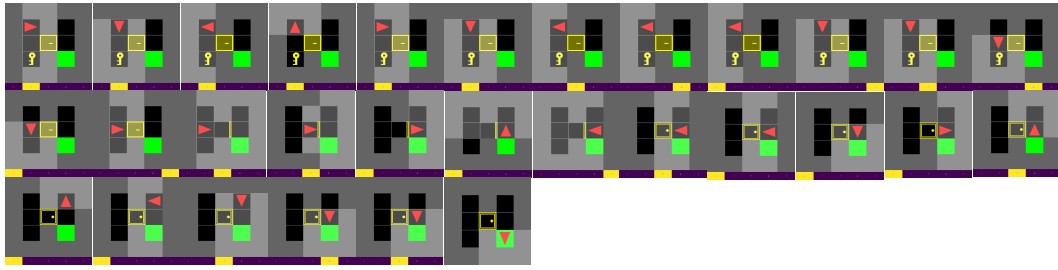

(c) Rollout 2: Doorkey, Agent without skip connections.

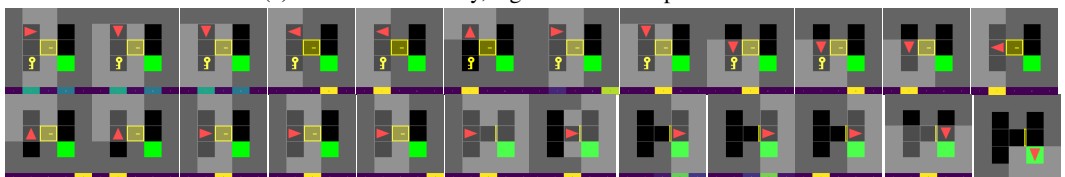

(d) Rollout 2: Doorkey, Agent with skip connections.

Figure 15: Trajectory rollouts (best viewed in a zig-zag fashion starting from top left) comparing agents with and without skip connections in the Doorkey environment. Each sequence shows one full trajectory.

