# OpenReview forum: "Handling Delay in Real-Time Reinforcement Learning"
_ICLR.cc/2025/Conference — ICLR 2025 Poster_

### Official Review · Reviewer_AMk5 · 2024-10-30

**Soundness:** 3
**Presentation:** 4
**Contribution:** 3
**Rating:** 8
**Confidence:** 3

**Summary:**

The authors explain the negative impacts of execution time and observation delay on real-time reinforcement learning performance. They describe a model architecture which includes “temporal skip connections” and “history augmentation”, explaining the theory behind it, and describing its impact on model expressiveness vs minimizing execution delay. Experimentally, they compare this new model architecture to existing architectures in several “game” environments, showing improved performance.

**Strengths:**

Very interesting paper:
- Great introduction to the problem space.
- Great theoretical motivation section.
- Architecture is well presented & Figure 2 is excellent.

**Weaknesses:**

Most of the paper speaks to the novel skip-connection architecture with history augmentation. As I understand the paper, some sections measure how existing parallelism approaches improve inference performance. I think these “mundane” sections take focus away from the novel content. Consider moving 1) the abstract sentence “accelerate inference by 6-350% on standard hardware” 2) Figure 1a, and 3) Section 5.4 to the appendix. This will help focus the paper on just the novel content. Apologies if the above is based on a misunderstanding of the paper content.

Table 2 would be easier to absorb if it included % deltas.

Erata:
- Typo: number of layes
- Inconsistent speech marks: As ”skip connections”

**Questions:**

I think the conclusion sentence “These architectures demonstrate robust performance across various environments and neuron execution times” undersells the paper results somewhat. Consider adding a slightly stronger follow-up statement “Experimentally, these architectures often perform better than traditional architectures” or similar.

---

> ### Author Response · Authors · 2024-11-20
>
> We thank the reviewer for the suggestions and positive evaluation of our work! We are excited that you found our paper interesting and that you appreciated our introduction to the problem, the theoretical motivation, and the presentation.

---

### Official Review · Reviewer_hZ2b · 2024-11-04

**Soundness:** 3
**Presentation:** 3
**Contribution:** 3
**Rating:** 6
**Confidence:** 4

**Summary:**

This paper presents a novel approach to addressing delays in real-time RL caused by neural network computation. The authors introduce a skip connection architecture that enables faster observation-to-action response by creating direct pathways in the network. This architecture is theoretically justified, and empirical results show its effectiveness over baseline models across multiple simulated environments. Through extensive ablation studies, the authors demonstrate that the observed performance improvements are due specifically to the skip connections, demonstrating the potential of this approach for real-time RL applications.

**Strengths:**

1. The authors introduce skip connections into neural networks to address inference delay in real-time RL. This approach is novel in the context of RL and represents the first application of skip connections specifically to reduce inference delay.
2. Extensive experiments validate the effectiveness of the proposed method. Results show that, with skip connections, agents operating under delay significantly close the performance gap compared to agents without delay and substantially outperform agents with delay but without skip connections.
3. The method is theoretically justified, and the theoretical findings are consistent with the experimental results.
4. The paper is clearly written and well-structured.

**Weaknesses:**

1. Experimental repetition is inconsistent across settings (10, 5, or 3 times), which could impact result reliability. Standardization or explanation is needed.
2. The neural network used is simple, resulting in minimal inference delay. Testing with more sophisticated models, such as computer vision-based architectures for processing observations, would better demonstrate real-world applicability. Additionally, the authors note that skip connections do not fully mitigate the performance drop from delay in complex tasks, raising the question: is this approach only effective for simple tasks where delay is less critical?
3. Scalability is limited, as larger architectures may encounter GPU memory constraints, reducing the benefits of parallel computation in complex environments.
4. Although targeting real-world delay issues, the method is only tested in simulations, leaving its real-world effectiveness unvalidated.

**Questions:**

N/A

---

> ### Author Response · Authors · 2024-11-20
>
> We thank the reviewers for the constructive comments and detailed feedback. The weaknesses are addressed as follows:
>
>
> > "Experimental repetition is inconsistent across settings (10, 5, or 3 times), which could impact result reliability. Standardization or explanation is needed."
>
> The different number of seeds across setups does not affect result reliability. The computed statistics (mean and standard error) automatically account for the varying number of seeds. While more seeds are always better for statistical robustness, we believe the current settings are sufficient for demonstrating the observed trends.
>
>
>  > "The neural network used is simple, resulting in minimal inference delay. Testing with more sophisticated models, such as computer vision-based architectures for processing observations, would better demonstrate real-world applicability."
>
> We will add this to the limitations section of the paper. However, we want to note that scaling vanilla RL methods to deeper neural networks is not straightforward and may include additional losses or training tricks. One can check the introduction of [1] for details. Since our focus is on using a basic algorithm, we chose not to explore these more complex scenarios.
>
> [1] Ceron, Johan Samir Obando, et al. "Mixtures of Experts Unlock Parameter Scaling for Deep RL." ICML2024
>
>
> > "...is this approach only effective for simple tasks where delay is less critical?"
>
> Please see point 3 in our common response for this.
>
> > "Scalability is limited, as larger architectures may encounter GPU memory constraints, reducing the benefits of parallel computation in complex environments."
>
> We do not fully understand why the proposed pipeline would encounter GPU memory issues with deeper architectures. During training, in the worst case with the naive implementation, the pipeline consumes a similar amount of memory as a recurrent neural network performing backpropagation through time, which is expensive but still manageable by modern hardware. During inference, memory usage should be even lighter. For example, on Figure 1a we estimates the speed of a 100-layer MLP on a single GPU with 40 GB memory.
>
> > "Although targeting real-world delay issues, the method is only tested in simulations, leaving its real-world effectiveness unvalidated."
>
> Please refer to the point 1 in the common response for this.

---

> > ### Comment · Reviewer_hZ2b · 2024-12-03
> >
> > Thanks for your response. I will keep my score.

---

### Official Review · Reviewer_3i3j · 2024-11-04

**Soundness:** 2
**Presentation:** 3
**Contribution:** 2
**Rating:** 6
**Confidence:** 3

**Summary:**

This paper mentions two main challenges in real-time reinforcement learning (RL): (1) limited action frequency due to neural network inference time, and (2) observational delay where actions are actually based on previous observations due to significant inference latency. While pipelining the policy network provides a framework to partially address these challenges, it is still not enough. The paper proposes two methods to overcome these challenges:
- Building on policy network pipelining, the paper introduces temporal skip connections which cache hidden representations from previous timestamps to compute the current action within a single layer's execution time. My understanding is that similar to residual connections, this approach can be viewed as an ensemble of multiple actors with different capabilities.
- The paper leverages history augmentation by incorporating previous actions to better maintain the Markovian property in real-time RL setting.
To validate their approach, this paper conducts experiments across various environments. The results demonstrate that when neuron execution time is small, the proposed method successfully accelerates inference without hurting performance. However, with longer neuron execution times or more complex environments, there is still a more noticeable degradation in performance.

**Strengths:**

- The paper did a good job presenting clear explanations of the motivation, research questions and statements, experimental methodology, and results. The writing is well-structured and easy to follow.
- The discussion of the actor expressive capacity vs observational delay is an interesting and important task in real-time reinforcement learning.
- The proposed methods could be motivated and connected with some theoretical understanding.

**Weaknesses:**

- The paper could potentially benefit from establishing a stronger connection between the theoretical analysis/motivation and experimental results. Specifically, including a concrete example that demonstrates how incorporating previous actions helps maintain the Markovian property which is broken before, and enhances training performance. This would strengthen the paper's arguments and make them more compelling.

- The proposed methods have suboptimal performance in certain environments, such as in MinAtar games. In my opinion, the paper could benefit from a more detailed/thorough discussion on those scenarios and possibly explore potential solutions, such as enhancing the action network's expressiveness to better handle the rapid changing observations (just a thought, not limit to this). These discussions/explorations would provide very valuable insights into current method's limitations and motivate future research.

- (Minor) The experimental evaluation could be further improved by including a broader range of testing environments, such as (could be a subset of) vanilla pixel-based Atari games. My understanding is that these experiments would further demonstrate the proposed methods are general across even more domains.

**Questions:**

- It would be helpful to provide clearer motivation and reasoning for how the neural execution times were chosen in different experiments. For example, providing both the environment step time and layer execution time would give readers a better sense of the magnitude of neural execution time in practice.

---

> ### Author Response · Authors · 2024-11-20
>
> We thank the reviewer for their constructive feedback and suggestions. We appropriate the reviewer's precise summary of our paper as well.  Below, we address each of your comments in detail:
>
> > "...Specifically, including a concrete example that demonstrates how incorporating previous actions helps maintain the Markovian property which is broken before, and enhances training performance. This would strengthen the paper's arguments and make them more compelling."
>
> We incorporated the example after Proposition 2 to provide the intuition. For qualitative analysis, please refer to Fig. 6. The agent without a skip connection relies on an outdated state for decision-making, effectively operating in a non-Markovian environment, which leads to wandering behavior.
>
> > "The proposed methods have suboptimal performance in certain environments, such as in MinAtar games. In my opinion, the paper could benefit from a more detailed/thorough discussion on those scenarios and possibly explore potential solutions, such as enhancing the action network's expressiveness to better handle the rapid changing observations (just a thought, not limit to this). These discussions/explorations would provide very valuable insights into current method's limitations and motivate future research."
>
> Please see point 3 in our common response for this.
>
>
> > "subset of vanilla pixel-based Atari games
>
> We have included experiments on a subset of Atari games in the Appendix B (page 13). These experiments demonstrate similar trends observed in the other environments discussed in the paper.
>
>
> > "It would be helpful to provide clearer motivation and reasoning for how the neural execution times were chosen in different experiments. For example, providing both the environment step time and layer execution time would give readers a better sense of the magnitude of neural execution time in practice."
>
> Since our experiments are conducted in simulations, the time to elapse one step of the environment depends on the hardware used. Therefore, we define neural execution times relative to the number of environment steps required to execute one layer.
>
> For instance, if the environment runs at 60 FPS, the environment step time will be $1/60$ seconds. If the neural execution time is set to 1, the execution of one layer also takes $1/60$ seconds. For an agent with a three-layer neural network and no skip connections, processing an observation into an action will take  $3/60$ seconds, but the agent outputs an action every  $1/60$ seconds anyway due to pipelining.
>
> For simplicity, we experiment with positive integer as neural execution times. This allows us simply to repeat the basic policy $\beta$ $\delta$ times while waiting for the agent’s action. Additionally, we vary this parameter to observe how performance changes as a function of neural execution time.
>
> We will expand on the motivation in the paper to make the experimental setup clearer.

---

### Official Review · Reviewer_aysZ · 2024-11-04

**Soundness:** 3
**Presentation:** 3
**Contribution:** 1
**Rating:** 5
**Confidence:** 3

**Summary:**

The paper proposes to use parallel pipeline computation and temporal skip connections to tackle real-time RL. It presents a formalization of the problem and proposed method as well as empirical results on Mujoco, MinAtar, and Minigrid.

**Strengths:**

- the paper is well structured and clearly written
- evaluation for both discrete and continuous action spaces
- error bars for quantitative analyses

**Weaknesses:**

- the novelty of the proposed method is limited to the (rather narrow) problem setting of real-time RL
- the empirical evaluation is limited to toy tasks (not really real-time RL problems); the paper would greatly benefit from a real-world application (or any application where the real-time constraint arises naturally)
- no reproducibility due to missing code

Especially the combination of the first two points restricts the contribution of this paper as both novelty and significance are severely compromised.

**Questions:**

- proposition 1 includes a relation between environment stochasticity and observational delay, i.e., the bound breaks down for deterministic environments; how do the empirical results relate to this?

---

> ### Author Response · Authors · 2024-11-20
>
> We thank the reviewer for the comments and constructive feedback. The weaknesses and questions are addressed as follows:
>
> > The empirical evaluation is limited to toy tasks...
>
> Please refer to our common response point 1 for a detailed discussion on this point.
>
> > The novelty of the proposed method is limited to the (rather narrow) problem setting of real-time RL.
>
> We respectfully disagree with this characterization. Real-time RL is widely regarded as an important and significant topic within the broader RL community, as evidenced by numerous papers [1,2,3,4] proposing solutions to this setting. These solutions often address focused subproblems of real-time RL that are not directly applicable to standard RL setups. In this context, we view our work as a meaningful contribution to the field, addressing challenges within real-time RL.
>
> [1] Travnik, Jaden B., et al. "Reactive reinforcement learning in asynchronous environments." Frontiers in Robotics and AI 5 (2018): 79.
>
> [2] Ramstedt, Simon and Christopher Joseph Pal. “Real-Time Reinforcement Learning.” NeurIPS2019.
>
> [3] Bouteiller, Yann, et al. "Reinforcement learning with random delays." ICLR2020.
>
> [4] Wang, Wei, et al. "Addressing Signal Delay in Deep Reinforcement Learning." ICLR2023.
>
>
> > no reproducibility due to missing code
>
> The code is not missing. The link to the code is provided in the footnote on page 5 of the original submission and on page 6 of the revised submission.
>
> > "Proposition 1 includes a relation between environment stochasticity and observational delay, i.e., the bound breaks down for deterministic environments; how do the empirical results relate to this?"
>
> While we did not conduct experiments in explicitly stochastic environments, the connection between stochasticity and observational delay remains relevant. As discussed in the paper after Proposition 2, even the deterministic environment appears stochastic to the agent due to stochasticity in its own policy, connecting it to the result of the Proposition 1.
>
> We also added experiments in explicitly stochastic environments by introducing "sticky actions" in MinAtar, where the agent's last action was repeated with a probability of $0.25$. Please, see Appendix B in the revised paper.

---

> > ### Comment · Reviewer_aysZ · 2024-11-25
> >
> > Thank you for your answer. I am sorry for having overlooked the code. I appreciate the reproducibility of your method and adjusted my score.
> >
> > I am not saying that real-time RL is insignificant or unimportant. Please do not misinterpret my comments. I am saying that the novelty is limited to real-time RL, i.e., the proposed methods have been used in other fields before. Since you did not object to that point I assume you agree with it.

---

> ### Author Response · Authors · 2024-11-27
>
> Thank you for adjusting the score and clarifying your concern regarding real-time RL.
>
> > I am saying that the novelty is limited to real-time RL, i.e., the proposed methods have been used in other fields before. Since you did not object to that point I assume you agree with it.
>
> We misunderstood your concern initially. You are correct that similar ideas have been applied before as noted in the related work section.  However, our contribution lies in integrating these ideas: pipelining, temporal skip connections from parallelization pipelines[1], training the critic without delay (since the critic is not used during inference), and augmentation with the last actions to recover the Markovian property from the delayed RL literature [2] -- into a method designed for real-time RL. Notably, the last action augmentation is enabled only through temporal skip connections, which is novel in our work as it emerges from the combination of pipelining, skip connections, and the real-time RL setting. We will emphasize the integration point in our revision.
>
> Secondly, the theoretical contributions in our paper are specific to the RL setting. Proposition 2 explicitly assumes access to actions, while Propositions 1 and 3 quantify performance gaps in terms of regret over the entire trajectory, rather than just future target predictions as in video or images. Moreover, our theoretical insights are not applicable to delayed MDPs, where the source of delay cannot be influenced. Therefore, the theory is specific to real-time RL and is not directly applicable to prior domains.
>
> Finally, we want to point out the small performance gap between the instantaneous (standard) and the real-time RL actors due to the combination of these techniques, and the alignment of our experiments with the theory.
>
>
> [1] Carreira, Joao, et al. "Massively parallel video networks." ECCV2018.
>
> [2] Wang, Wei, et al. "Addressing Signal Delay in Deep Reinforcement Learning." ICLR2023.

---

> > ### Author Response · Authors · 2024-12-02
> >
> > Thanks again for engaging with our work.  Did our response address your concerns with novelty?  Can you please raise any remaining concerns, questions, or feedback before the end of the discussion phase in 2 days on Dec 3, so we have time to respond further?

---

### Author Response · Authors · 2024-11-20
**Rebuttal**

We thank the reviewers for the constructive comments and feedback. The reviewers raised several common concerns, which we address below.

**1. Real-world application (Reviewers aysZ, hZ2b):**

    "...the paper would greatly benefit from a real-world application...", "...targeting real-world delay issues, the method is only tested in simulations..."

Unfortunately, because we lack access to robots, our experiments must be confined to what is possible with simulators. We consider Mujoco to be a simulation platform for robotics and have already conducted experiments using it. If you have any additional suggestions regarding real-time environments, we would be grateful to hear them.

We also would like to emphasize that real-world constraints are naturally embedded in the simulations or games we consider. These games simulate scenarios similar to how humans interact with real-world systems. For example, in Breakout, humans play at a fixed frame rate, and the ball does not pause for human input before continuing its movement. As a result, our agent's behavior directly mirrors how humans play games on an actual Atari console in the real-world.

Moreover, we added a subset of Atari games in Appendix B, as requested by Reviewer 3i3j, that use a more complex pixel based observation space.  We believe our experiments in diverse environments, including Mujoco, MinAtar, MiniGrid and subset of Atari games, are sufficient to demonstrate the advantages of the proposed pipeline in real-time scenarios and how it compares to instantaneous actor.

**2. Connecting theory and experiments (Reviewers aysZ, 3i3j):**

    "...the bound breaks down for deterministic environments; how do the empirical results relate to this?...", "...establishing a stronger connection between the theoretical analysis/motivation and experimental results..."

We have addressed these concerns by incorporating Reviewer 3i3j's suggestion to add a motivated example for Proposition 2 into the paper. We also have emphasized the connection between Proposition 1 and experiments after Proposition 2, using italicized text.

Finally, we added new paragraph at the end of Section 4.2 connecting performance gap between instantaneous and real-time actors to the theory, that we include here as well:

Propositions 1, 2 highlight the performance gap between agents with and without skip connections and action augmentation, in terms of delay regret. Besides, Propositions 1 and 3 (Appendix H) provide insights into the performance gap between the instantaneous and real-time actors in a parallel computation framework under worst-case scenarios. Notably, even with skip connections, the delay $\delta$ remains. In contrast, the instantaneous actor does not experience any delay or inaction regrets. The closer the environment is to a worst-case scenario, the more pronounced the performance gap becomes.

The new text is highlighted in red.

**3. Discussion on the proposed approach's effectiveness and limitations (Reviewers 3i3j, hZ2b):**

    "...The proposed methods have suboptimal performance in certain environments ... the paper could benefit from more discussion on those scenarios...", "...skip connections do not fully mitigate the performance drop in complex tasks...is this approach only effective for simple tasks where delay is less critical?"

We agree that additional discussion would enhance the paper. As suggested by Reviewer 3i3j, one potential solution could involve increasing the neural network's expressivity or reducing the neural execution time (delay) by merging multiple layers into a single basic block. However, if the architecture and its associated delay are fixed, the optimal solution achievable with this architecture may be strictly worse compared to an instantaneous actor, as supported by our distillation experiment (Section 5.3) and Proposition 1, 3.

We added this discussion at two places: highlighting theoretical performance gap between between instantaneous and real-time actors at the end of Section 4.2 and at the end of Section 5.1 while discussing experimental results. The new text is colored in red.

Additionally, we believe that further theoretical development is needed to quantify the performance gap more precisely, especially when considering the "complexity" of the environment.

---

### Meta-Review · Area_Chair_279E · 2024-12-21

**Metareview:**

This paper presents methods to more efficiently train real-time RL systems by leveraging parallel computations of the layers. The main idea is to address the observational delay. The proposed method is theoretically justified and empirical results from simulations demonstrate the effectiveness of the approach. While the proposed techniques have been explored in other areas, the contribution to the RL seems sufficient, e.g., introducing a set of new techniques to a broader audience has some value. Overall, the paper is well written and most reviewers have a positive view of the paper. The paper could be strengthened if experiments were conducted using real-world applications.

**Additional Comments On Reviewer Discussion:**

The authors did address the reviewers' concerns during the rebuttal period. The remaining concerns are the novelty/contribution of this paper, specifically, regarding how "novel" is the approach and the missing real-world application. The AC felt that having real-world applications will be convincing, however, the authors explained that they do not have the hardware to do so. Overall, the strengths of the paper marginally outweigh the weaknesses, and hence the AC is recommending an acceptance.

---

### Decision · Program_Chairs · 2025-01-22

Accept (Poster)